# Characterization of Degraded Konjac Glucomannan from an Isolated *Bacillus licheniformis* Strain with Multi-Enzyme Synergetic Action

**DOI:** 10.3390/foods13132041

**Published:** 2024-06-27

**Authors:** Xueting Zhang, Jieqiong Ding, Minghong Liao, Xin Meng, Yubiao Fu, Linjuan Huang, Zhongfu Wang, Qingling Wang

**Affiliations:** Shaanxi Natural Carbohydrate Resource Engineering Research Center, College of Food Science and Technology, Northwest University, Xi’an 710069, China; 18846152669@163.com (X.Z.); 202323969@stumail.nwu.edu.cn (J.D.); 202233799@stumail.nwu.edu.cn (M.L.); 2021120010@stumail.nwu.edu.cn (X.M.); 2022120029@stumail.nwu.edu.cn (Y.F.); huanglj@nwu.edu.cn (L.H.); wangzhf@nwu.edu.cn (Z.W.)

**Keywords:** microbial degradation, konjac glucomannan, food spoilage, multi-enzyme synergetic action

## Abstract

The large molecular weight and high viscosity of natural konjac glucomannan (KGM) limit its industrial application. Microbial degradation of low-molecular-weight KGM has health benefits and various biological functions; however, the available KGM strains used in the industry have microbial contamination and low degradation efficiencies. Therefore, exploring novelly adaptable strains is critical for industrial processes. Here, the *Bacillus licheniformis* Z7-1 strain isolated from decaying konjac showed high efficiency for KGM degradation. The monosaccharide composition of the degradation products had a reduced molar ratio of mannose to glucose, indicating that Z7-1 preferentially degraded glucose in KGM. The degraded component was further characterized by ESI-MS, Fourier-transform infrared spectroscopy (FT-IR), and scanning electron microscopy (SEM), and it also exhibited good antibacterial activity against various food-spoilage bacteria. Genome sequencing and zymolytic analysis revealed that abundant carbohydrate-active enzymes exist in the Z7-1 genome, with at least five types of extracellular enzymes responsible for KGM degradation, manifesting multi-enzyme synergetic action. The extracellular enzymes had significant thermal stability, indicating their potential application in industry. This study provides an alternative method for obtaining low-molecular-weight KGM with antibacterial functions and supports foundational knowledge for its development as a biocatalyst for the direct conversion of biomass polysaccharides into functional components.

## 1. Introduction

Konjac glucomannan (KGM), a neutral water-soluble heteropolysaccharide extracted from the tuber of *Amorphophallus konjac* C. Koch, has been generally used in the food and medical industries because of its unique physicochemical properties [1,2,3]. KGM is composed of *β*-1,4-linked D-glucose and D-mannose in 1:1.6 or 1:1.4 molar ratios with traces of acetyl groups randomly present at C6 and several short mannose branches at C3 positions [4,5]. KGM is widely used in the functional food and medicine industries because of its biological functions, including directing the microbiota community, immunomodulation, anti-obesity and anti-diabetic properties, wound healing, and preventive effects on cancer [1,6,7,8], indicating that KGM has considerable application potential as a functional food. However, its large molecular weight and high viscosity affect its utilization in the intestinal tract and industrial applications, resulting in a considerable reduction in its overall utilization efficacy.

Compared to natural KGM, degraded KGM has desired gut health benefits, promotes probiotic growth, adsorbs pathogenic bacteria, lowers blood lipids, and scavenges free radicals, and these properties have attracted interest in the food industry [5,9,10,11]. Low-molecular-weight KGM is typically obtained using physical, chemical, and biological methods. The biological methods can include microbiological and enzymatic catalysis, which have considerably wider applications owing to their environmental friendliness, high efficacy, simple process, energy efficiency, and reproducibility [12,13,14]. Currently, enzymatic hydrolysis is widely used for the degradation of polysaccharides due to characteristics such as safety and high selectivity [15,16,17]; however, it is difficult to obtain a large amount of low-molecular-mass KGM or oligosaccharides by enzymatic hydrolysis alone because it is an expensive process. One way to overcome this limitation is to develop microbial catalysts with efficient KGM degradation activity.

Currently, various KGM-degrading microorganisms are used to obtain low-molecular-weight KGM components or oligosaccharides [14,18,19], with cellulases and *β*-mannanases being two major carbohydrate-active enzymes responsible for KGM degradation [12,13]. *β*-mannanases hydrolyze the *β*-D-1,4 linkages between mannose residues [20,21], while cellulase hydrolyzes *β*-D-1,4 linkages of glucose in the polysaccharide backbone of glucomannan and galactomannan, producing low-molecular-weight manno-oligosaccharides [22]. *Aspergillus niger* is an important industrial fermentation strain that can produce *β*-mannanase which is widely used in the degradation of KGM. The addition of 2% KGM substrate to the culture of *A. niger* resulted in a 32.2% hydrolysis efficiency of natural KGM to oligosaccharides [19]. *Syn. Hypocrea jecorina* is currently one of the main industrial sources of glycoside hydrolases. Mikkelson et al. analyzed glucomannan obtained from the hydrolysis of KGM using two types of endocellulases (Tr Cel5A and Tr Cel7B) and one type of mannanase (glycoside hydrolase family 5) [12]. With a substrate concentration of 1.0% konjac flour in the reaction system, the oligosaccharide yield was 17% and resulted in a polymerization degree of 2–6 [12]. In addition, strains such as *Bacillus subtilis*, *Bacillus amyloliquefaciens*, and *Aspergillus oryzae* can depolymerize KGM into oligosaccharides or low-molecular-weight KGM, with most substrate concentrations controlled at 1–2% during the degradation process [14,18,23].

Although these microorganisms show good performance in the degradation of konjac polysaccharides, some limitations remain. For example, these strains can only degrade low-concentration konjac flour, which is usually controlled at 1%, and cannot hydrolyze high-concentration KGM, which limits their application in industrial production [12,13]. The products degraded by these strains are also contaminated with other bacteria, which affects the quality of the degradation products. In addition, industrial strains require extreme environmental conditions during the degradation process, such as high temperatures, which further limits their application in practical processes. Given these limitations, screening for KGM-degrading strains that have high efficiency, are safe, and have adaptability is of great significance for the development and utilization of konjac flour.

Here, one strain with highly efficient KGM degradation ability was isolated from decaying konjac and identified as *Bacillus licheniformis* Z7-1 based on 16S DNA, microscopy, and morphology. The degradation ability of strain Z7-1 at different concentrations of KGM substrate was determined using viscosity analysis. Structural and biochemical properties were determined using thin-layer chromatography (TLC), ESI-MS, FT-IR, and SEM. The antibacterial activities of the degraded KGM components against various food-spoilage bacteria and the carbohydrate-active enzymes of strain Z7-1 were investigated, along with the stability of the extracellular enzymes. This study provides an alternative method for obtaining low-molecular-weight KGM components with antibacterial activity and demonstrates their application in converting natural konjac biomass into functional components.

## 2. Materials and Methods

### 2.1. Materials and Chemicals

Konjac glucomannan was purchased from Hubei Konson Konjac Technology Co. Ltd. ((KJ-30, Hubei Qiangsen Konjac Technology, Ezhou, China). *Bacillus cereus*, *Pseudomonas fragi*, and *Staphylococcus aureus* were obtained from refrigerated meat and seafood samples in our laboratory. Analytical-grade 3,5-dinitrosalicylic acid (DNS), *N*-butanol, glacial acetic acid, trifluoroacetic acid, sodium borohydride, methanol, pyridine, acetic anhydride, ethanol, and acetonitrile were purchased from Jingbo Biotechnology Co., Ltd. (Taiyuan, China). D-mannose, glucose, galactose, and maltodextrin were purchased from Sigma-Aldrich (Shanghai, China).

### 2.2. Isolation and Identification of KGM-Degrading Bacteria

The KGM-degrading bacteria were screened from decaying konjac using the microbial enrichment method combined with Congo red staining with the addition of a certain amount of KGM (*w*/*v*, 0.5%) into Luria–Bertani agar plates. The target strain with excellent KGM degradation ability was evaluated using the size of the hydrolysis circle, TLC, and viscosity analysis. The colony morphology and Gram stain were observed using an optical microscope after incubation for 12 h at 37 °C. The genomic DNA of the strain was extracted and the 16S rDNA gene was cloned by PCR with the forward primer 5′-AGAGTTTGATCMTGGCTCAG-3′ and the reverse primer 5′-TACGGYTACCTTGTTACGACTT-3′. The amplified DNA fragments were sequenced by Tsingke Biotechnology Co., Ltd. (Beijing, China). A phylogenetic tree based on the 16S rDNA gene sequences was constructed using MEGA 7.0 software.

### 2.3. TLC Analysis of KGM Hydrolysis Products

Thin-layer chromatography analysis of the KGM hydrolysates was performed on silica gel glass plates. A solution of *N*-butanol, glacial acetic acid, and water in a 3:3:2 ratio (*v*:*v*:*v*) was used as a developer for good separation; maltodextrin was used as the standard, natural KGM (*w*/*v*, 1%) was used as the control, and KGM degradation products produced by strain Z7-1 were used as the samples. Sample aliquots of 2 µL were placed at the starting line and then blown dry. This process was repeated five times. The spots were visualized by spraying samples onto silica gel glass plates with the colorants and incubating them at 105 °C for 20 min. The colorants were prepared by dissolving 0.5 g of diphenylamine, 0.5 g of aniline, 25 mL of trichloroacetic acid, and 25 mL of acetone. 

### 2.4. Growth and Enzyme Production 

Growth curves were determined from strain Z7-1 cultivated at 37 °C with 180 rpm/min shaking. Bacterial samples were collected every 12 h and detected at OD_600_. Enzymatic activity was determined as follows: First, 200 µL of enzyme preparation was mixed with 500 µL of 0.5% KGM in 20 mM PBS and incubated for 30 min at 55 °C. Then, 1 mL of DNS was added to the reaction system and incubated for an additional 10 min. After cooling the reaction mixture to room temperature, it was centrifuged for 10 min at 12,000× *g*, and the supernatant was diluted to 10 mL. The absorbance of 150 µL of the diluted supernatant was measured at 540 nm. A standard curve for mannose was generated using a series of mannose concentrations. The amount of enzyme required to generate 1 µM mannose per minute was defined as one enzyme activity unit (U), and mannanase activity was calculated as (*V*/mL) = *A* × *N*/T where *A* is the amount of mannose by enzyme activity (μmol), *N* is the dilution ratio of enzyme solution, and *T* is the enzyme reactive time (min).

### 2.5. Determination of KGM Degradation Ability of Z7-1

Strain Z7-1 obtained was inoculated into 20 mL of Luria–Bertani broth medium in a 50 mL conical flask and cultured at 37 °C for 12 h with shaking at 180 rpm. Then, 2 mL of the culture was transferred into 100 mL of Luria–Bertani broth medium and then was cultured for 48 h at 37 °C before different concentrations of 5%, 10%, 15%, and 20% konjac flour were added to the cultures. Samples of the bacterial hydrolysis products were collected at 12 h and 24 h, and the changes in viscosity were measured using an Anton Paar MCR 302 rheometer. A power-law model was used to fit the viscosity of each sample to the shear rate: τ = K × γ n, where τ represents shear stress, K is the consistency coefficient or power-law coefficient (Pa⋅sn), and n represents the fluidity index or power-law index (unitless). The K value is a measure of viscosity but is not equal to the viscosity value. The higher the viscosity, the higher the K value.

### 2.6. Physicochemical Characterization of the KGM Degradation Components 

#### 2.6.1. Determination of Molecular Weight and Monosaccharide Composition

The degradation products were centrifuged at 12,000× *g* for 30 min, and the supernatant was collected, evaporated, and freeze-dried, from which 2 mg was dissolved in 20 mM phosphate buffer solution (pH 6.0) and then filtered through a 0.22 μM membrane to remove impurities for further analysis. The molecular masses of the partially degraded KGM were determined using high-performance liquid gel permeation chromatography (HPGPC). Twenty microliters of the sample was injected into a TSKgel G4000PWXL (300 × 7.8 mm, 10 μm, Beijing Green Herbs Science and Technology, Beijing, China) dextran gel chromatography column. The mobile phase contained phosphate buffer (20 mM, pH 6.0) with a flow rate of 0.3 mL/min. 

Monosaccharide composition analysis of the degraded KGM components was performed as previously described [24]. Four milligrams of the sample was hydrolyzed by trifluoroacetic acid at 121 °C for 2 h in a reaction flask, and then the pH of the hydrolysate was adjusted to neutral. NaBH_4_ (*w*/*v*, 4%, 0.5 mL) was added and allowed to react at room temperature for 1.5 h. Excess NaBH_4_ was removed by adding glacial acetic acid and methanol. Then, 1 mL of pyridine and acetic anhydride was added into the reaction system to perform the acetylation, followed by incubation overnight at room temperature. The targeted sample was dried under vacuum at 80 °C and further extracted using dichloromethane and double-distilled water.

#### 2.6.2. ESI-MS Analysis

The cells were removed by centrifugation at 12,000× *g* for 30 min, and the degraded KGM oligosaccharides were purified using a Sep-Pak C18 column (250 mg/3 mL; Simon Aldrich, Germany) combined with a graphite carbon solid-phase exchange column (250 mg/3 mL; Nantong Hai Ruo Chemistry Technology, Nantong, China). The targeted oligosaccharides were dried under a stream of nitrogen at room temperature. The identification and analysis of the KGM oligosaccharides were performed using ESI-MS (LTQ-Tune; Thermo Scientific, Waltham, MA, USA) as follows: A volume of 2 μL of the sample solution was injected into the electrospray ion source by a stream of 50% methanol at a flow rate of 20 μL·min^−1^ from the pump of HPLC system. For the electrospray ion source, the spray voltage was set at 4 kV, with a sheath nitrogen flow rate of 20 arb, an auxiliary gas (nitrogen) flow rate of 10 arb, temperature at 300 °C, voltage at 37 V, and a tube lens voltage of 250 V. 

#### 2.6.3. FT-IR and SEM

The structure of the KGM degradation products was determined using FT-IR and SEM. Cells were removed by centrifugation at 12,000× *g* for 30 min, and the supernatant was sedimented by sequentially incubating the sample with a 30%, 50%, and 70% ethanol concentration series for 24 h. Then, the precipitate from the 30%, 50%, and 70% ethanol concentrations and the supernatant after 70% precipitation were collected, evaporated, and freeze-dried. The component from 70% ethanol precipitate was used for further FT-IR and SEM analysis. 

Dried degraded KGM samples (2 mg) were mixed with KBr powder and pressed into pellets. The KBr pellets were scanned with an FT-IR spectrometer (Nicolet 5700, Thermo Fisher Scientific, Madison, WI, USA) over a range of 4000 to 400 cm^−1^, with a resolution of 4 cm ^−1^. Dried samples were applied to double-sided conductive tape and sprayed with gold for 15 min. The morphologies of the samples were observed using a JMS-6701 F Field Emission SEM (Japan Electronics Co., Ltd., Tokyo, Japan) at an accelerating voltage of 3.00 kV and magnifications of 250× and 1000×.

#### 2.6.4. Antibacterial Activities 

The antibacterial effects of KGM degradation components against various food-spoilage bacteria, including *B. cereus, P. fragi*, and *S. aureus* were evaluated using the Oxford cup method. The Z7-1 strain was cultured for 48 h at 37 °C, and then the 5% konjac flour was added to the cultures, followed by further degradation for 24 h. Cells were removed by centrifugation at 12,000× *g* for 10 min, and the supernatant samples were used for the antibacterial activity assays. An Oxford cup (inner diameter, 7.8 mm; Xuzhou Xinri Technology, Xuzhou, China) was filled with the supernatant samples at a volume of 20, 50, 100, and 150 µL, and 150 µL of sterile water was used as the control. Then, the set-up was incubated at 37 °C for 36 h. The evaluation of antimicrobial activities was performed based on clear zones developed around the wells. 

### 2.7. Thermal Stability of Extracellular Enzymes and Zymogram Analysis

The thermal stability of the extracellular enzymes was evaluated by incubating the enzyme sample at 55 °C and 80 °C for 0.5–4 h. The residual activity was determined at 55 °C and pH 8.1. Zymography was conducted using a 12% polyacrylamide separating gel containing 0.5% KGM as a substrate. After electrophoresis, the gel was washed twice in 2.5% (*v*/*v*) Triton X-100 for 30 min at room temperature to remove the SDS, rinsed three times with distilled water, and incubated in reaction buffer (50 mmol/L Tris-HCl pH 8.3, 50 mmol/L CaCl_2_) at 37 °C for 2 h. The gels were stained with 1 mg/mL Congo red solution and washed with tap water three times, and then 1 M NaCl was used to wash the Congo red until transparent stripes were observed.

### 2.8. Carbohydrate-Active Enzymes of Strain Z7-1 for KGM Hydrolysis

The whole gene framework of Z7-1 was sequenced by second-generation sequencing (Qingke Biotechnology Co., Ltd., Beijing, China). Prokka v1.14.5 software was used to annotate protein and non-coding genes, such as tRNA and rRNA, in the Z7-1 genome. EggNog mapper v2.1.5 software was used for the functional annotation of protein-coding genes. The CAZy enzyme database was used to analyze the carbohydrate-active enzymes in strain Z7-1, including glycoside hydrolases (GHs), glycosyltransferases (GTs), polysaccharide lyases (PLs), carbohydrate esterases (CEs), auxiliary oxidoreductases (AAs), and carbohydrate-binding modules (CBMs), with or without catalytic activity. Multiple enzymes (e.g., cellulase, mannanase, α-galactosidase, lytic polysaccharide monooxygenases, deacetylase) responsible for KGM degradation were also analyzed using a carbohydrate-active enzyme database.

### 2.9. Statistical Analysis 

Each experiment was repeated at least in triplicate. The data are expressed as the mean ± standard deviation (SD). Statistical analysis of the data was performed using one-way analysis of variance (ANOVA) with SPSS Software (v15.0, SPSS Inc., Chicago, IL, USA). Differences were considered significant at *p*-values < 0.05. Origin software (version 9.0) was used to generate graphs.

## 3. Results

### 3.1. Isolation and Identification of the KGM-Degrading Strain

Among the strains screened from decaying konjac, strain Z7-1 displayed the best KGM degradation ability. Strain Z7-1 formed a white, moist, and circular colony with a wrinkled surface on LB agar plating medium containing 0.5% KGM and generated an obvious circle around the colony with Congo staining (Figure 1a). TLC analysis revealed that KGM was decomposed gradually into a mixture of small oligosaccharides and low-molecular-weight KGM in the strain Z7-1 cultures (Figure 1b). PCR amplification of the 16S rDNA gene from strain Z7-1 was conducted and compared with those of other strains using the BLAST program in the NCBI database. A phylogenetic tree that included the Z7-1 and reference strains of each species with neighbor-joining methods was constructed (Figure 1c). The tree showed that strain Z7-1 was mostly associated with *Bacillus licheniformis* O5, with homology >99%. 

Kinetic curves of enzymatic production indicated that strain Z7-1 reached a stable enzyme activity after culture for 24 h with no obvious lag period, and could maintain high activity even after 144 h of cultivation (Figure 1d). Growth curves indicated that strain Z7-1 could reach the stable period after nearly 72 h of culture and did not reveal a decline even after 144 h (Figure 1e). These properties of strain Z7-1 suggest its potential application in industrial processes.

### 3.2. Degrading Ability of Strain Z7-1 for Substrates with Different KGM Concentrations

In this experiment, strain Z7-1 was cultivated in LB for 48 h at 37 °C before 5%, 10%, 15%, or 20% konjac powder was added to the cultures. The degraded products were then measured sequentially at 12 h and 24 h using apparent viscosity analysis (Figure 2). Strain Z7-1 showed excellent degradation activity for a 5% substrate concentration with a significant reduction in apparent viscosity after hydrolysis for 12 h (Figure 2a). When the KGM concentration was increased to 10% or 15%, the viscosity of the product exhibited an obvious reduction, with a stronger trend observed at 24 h compared to that at 12 h (Figure 2b,c). Although no significant viscosity change was observed at 12 h towards 20% KGM substrates, after 24 h, an obvious viscosity reduction was observed when compared to that of the control group (Figure 2d), indicating that strain Z7-1 still maintains its efficient degradation ability towards high-concentration KGM substrates. 

### 3.3. Characterization and Purification of KGM Degradation Products 

The average molecular weights of degraded products were determined using the HPGPC method. The samples were taken at different time intervals (12 h, 24 h, 48 h, 96 h, and 144 h) when the KGM substrate concentration was controlled at 5%. Cells were centrifuged at 12,000× *g* for 30 min, and the supernatant was collected, evaporated, and freeze-dried. The samples were dissolved in 20 mM PBS pH = 6.0 and were further assessed by HPGPC analysis. As shown in Figure 3a, after degradation of 12 h or 24 h, the degraded products (4.17 ± 0.05 kDa for 12 h and 4.06 ± 0.09 kDa for 24 h) presented wider peaks in the GPC profile, indicating that the components are probably composed of a series of polymers with different molecular weights. When the time was extended to 48 h (4.01 ± 0.07 kDa), 96 h (4.05 ± 0.02 kDa), and 144 h (4.35 ± 0.08 kDa), the molecular weight of degraded products was distributed around 4 kDa, generating a more homogeneous KGM component compared to that of 12 h or 24 h products (Figure 3a).

To analyze the components of the KGM oligosaccharides generated after the degradation by *B. licheniformis* Z7-1, the degradation products were measured by ESI-MS using the observed mass-to-charge (*m*/*z*) values of the ions (Figure 3b,c). ESI-MS analysis of the reaction mixtures confirmed that *B. licheniformis* Z7-1 generated a series of native KGM oligosaccharides and a small amount of corresponding oxidized KGM oligosaccharides. The oligosaccharides produced by strain Z7-1 after 12 h of hydrolysis had a wide distribution of polymerization degrees (Figure 3b; DP2-DP9). After 144 h, the polymerization degrees of oligosaccharides were mainly DP6-DP7 (Figure 3c), indicating strain Z7-1 prefers to use the DP2-DP5 oligosaccharides during degradation. Taking the dimer for example, possible products in ion clusters were mostly the native KGM oligosaccharide dimer ([M + Na]^+^ = *m*/*z* 365.08), the C1-oxidized aldonic acid or C4-oxidized gemdiol (hydrated species, [M + Na]^+^ = *m*/*z* 381.08), or the K^+^ signal fragment ([M + K]^+^ = *m*/*z* 381.08), and the native dimer oligomers with a signal acetyl group ([M + Na + acetyl]^+^ = *m*/*z* 407.17). 

The monosaccharide compositions of degraded components were determined using gas chromatography. The ratios of mannose to glucose, which were 1.46 at 12 h and 1.32 at 144 h for 5% KGM substrate, exhibited a reduction trend during the degradation process; meanwhile, a similar trend was observed with a ratio of 1.67 at 12 h and 1.33 at 144 h for 1% KGM substrate; these results indicate that strain Z7-1 preferentially degrades glucose in KGM (Figure 3d).

Food spoilage is a major concern in the food industry and can cause serious harm to human health. In this study, the antibacterial properties of the KGM degradation products were determined by evaluating their ability to inhibit the growth of *B. cereus, P. fragi, and S. aureus*, which are common microbes found in food that can pose severe health risks. When 150, 100, 50, and 20 µL of degraded products were added into the holes of an Oxford cup, obvious antibacterial zones on the LB solid agar plates were observed, and the diameters of the clear zones were recorded. Larger antibacterial circles could be observed with the addition of more KGM degradation products into the cups (Figure 3e), and the diameters of clear zones were as follows: *B. cereus* (a, 8.01 ± 0.02 mm; b, 8.53 ± 0.01 mm; c, 9.01 ± 0.04 mm; d, 9.24 ± 0.05 mm), *P. fragi* (a, 7.90 ± 0.03 mm; b, 8.61 ± 0.03 mm; c, 9.51 ± 0.08 mm; d, 9.98 ± 0.02 mm), *S. aureus* (a, 8.01 ± 0.03 mm; b, 8.63 ± 0.01 mm; c, 9.60 ± 0.10 mm; d, 10.18 ± 0.05 mm). These results indicated that the KGM products degraded by strain Z7-1 exhibited antibacterial activity against various food-spoilage bacteria, implying their application in food preservation.

To obtain highly uniform KGM degradation components, the degradation products were immersed in ethyl alcohol with different concentrations series for 24 h after hydrolysis for 48 h by strain Z7-1. Compared with the components from 30% and 50% alcohol precipitation, the elution peaks of the 70% component detected via HPGPC had better symmetry, suggesting that the composition of the 70% component was more homogeneous than that of the 30% or 50% component (Figure 4a). Compared to that of 30% (DP2-DP6) and 50% (DP2-DP7) alcohol precipitations, as well as the supernatant after 70% alcohol precipitation (DP2-DP6) (Figure 4b–d), there was a wider range of KGM oligosaccharides, mainly DP2-DP10, in the 70% alcohol precipitation (Figure 5a) based on the ESI-MS analysis.

FT-IR characterization was performed to further investigate the degradation of KGM by strain Z7-1. The FT-IR spectra that corresponded to the molecular vibrations of the samples showed that the intensities of the absorption peaks varied slightly for the purified samples (Figure 5b). No new chemical groups were introduced into the KGM molecular chain after degradation, with the main bands for KGM observed at around 4000–800 cm^−1^. The peaks at 3384 cm^−1^ and 2927 cm^−1^ were assigned to the stretching vibrations of the O–H groups and methyl C-H groups [24,25]. The peak at about 1647 cm^−1^ was attributed to the C=O in the acetyl group [26]. All the samples exhibited flexural vibration peaks for C–H at 1411 cm^−1^ and C–O at 1155 cm^−1^ [27]. This result showed that strain Z7-1 did not remove the chemical groups in KGM during degradation.

SEM was used to visualize KGM sample morphologies (Figure 5c). The untreated KGM exhibited a flake-type structure with diverse particle sizes and shapes. After degradation by strain Z7-1, the particles disappeared and the surface became porous. These pores may have been caused by the breaking of the glycoside bond during the degradation process. The morphology clearly showed that the samples degraded by strain Z7-1 were less dense and more porous and had an irregular sheet-like surface structure. These results indicated that konjac powder was degraded by strain Z7-1 at both 1% and 5% concentrations. 

### 3.4. Carbohydrate-Active Enzymes of Strain Z7-1 for Hydrolysis of KGM

Cluster of orthologous group (COG) analysis identified 4876 gene orthologs (GOs), 2550 (84%) of which were successfully assigned to known protein-coding GOs, while the rest were assigned to genes of general or unknown function (Figure 6a). A total of 1408 GOs were assigned to the category “metabolism”, most being in the subcategories amino acid transport and metabolism (298 GOs), energy production (182 GOs), and carbohydrate transport and metabolism (260 GOs). A total of 671 GOs were assigned to the category “information storage and processing”, most of which were assigned to the subcategories translation (164); DNA replication, recombination, and repair (135); and transcription (275). The “cellular process and signaling” category was dominated by GOs involved in post-translational modification, protein turnover, chaperones (108), and cell wall/membrane/envelope biogenesis (187).

Seven gene copies each of CE4 and CE9 CAZymes that catalyze the deacylation of polysaccharides are present in the genome of Z7-1, which is consistent with the ESI-MS results that showed a low degree of acetylation in KGM oligosaccharides degraded by Z7-1 (Figure 3b,c and Figure 5a). GHs with an active site for these polysaccharides were also identified, with the highest number of subfamilies and gene copies for those targeting cellulase activity (5), mannanases (2), and α-galactosidase (1) that are responsible for KGM degradation (Figure 6c,d). Previously, chitin-active bacteria-derived lytic polysaccharide monooxygenase (LPMO) *Sm*AA10A was used to catalyze KGM degradation and generate diverse oxidation products including single C1, single C4, and C1/C4 doubled sites [24]. A coding protein annotated with probable chitin-active LPMO existed in the genome of strain Z7-1, suggesting its potential degrading activity for KGM. Zymolytic analysis revealed that at least five extracellular enzymes, including three major and two minor enzymes, participate in KGM degradation (Figure 6e).

### 3.5. Thermal Stability of the Extracellular Enzymes for KGM Hydrolysis

Enzyme stability is critical for industrial biomass degradation applications. The thermal stability of extracellular enzymes responsible for KGM degradation was assessed and showed high stability at high temperatures, retaining about 91–108% of the initial activity at 55 °C after incubation for 0–4 h, and maintaining 70% of the initial activity after 0.5 h incubation at 80 °C, which was approximately 60% of the residual activity retained after 4 h incubation (Figure 7). This result suggested that the extracellular enzymes of strain Z7-1 have potential for future industrial applications that require enzyme stability under conditions of high temperatures.

## 4. Discussion

Owing to the advantages of biocompatibility, efficiency, and environmental friendliness, the biodegradation of polysaccharides has tremendous potential for industrial applications [14,28,29,30]. Among these biodegradation processes, microbiological biocatalysts can secrete efficient enzymes for the direct conversion of polysaccharides into oligosaccharides or low-molecular-weight polysaccharides, demonstrating their advantages over the addition of relatively expensive enzymes for substrate degradation [14,18,31]. KGM has attracted increasing research and industrial interest because of its particular biological functions as a food source with various industrial applications, as well as the production of low-molecular-weight components or oligosaccharides upon its degradation [30,32,33]. Although several microbiological biocatalysts have been reported to depolymerize natural KGM into low-molecular-mass components or oligosaccharides, such as *Aspergillus oryzae* and *Trichoderma reesei* [12,23], easy contamination by other bacteria, low degradation efficiency, and only acting on low-concentration KGM substrates limit the wide application of these strains in the industry. Here, strain Z7-1 was isolated from a screen using decaying konjac and identified as *Bacillus licheniformis*. The viscosity for products degraded by strain Z7-1 towards different KGM substrate concentrations was determined to evaluate the degradation ability. The functional and physicochemical properties of the components degraded by Z7-1 were determined, and the carbohydrate-active enzymes responsible for KGM degradation were analyzed using genome sequence analysis to elucidate the potential mechanism of the high degradability of Z7-1. The excellent thermal stability of the extracellular enzymes suggests the potential industrial application of Z7-1. 

The target strain was obtained based on the size of the hydrolyzed circle, TLC analysis, and viscosity changes of the KGM substrate, and it was identified as *Bacillus licheniformis* and named Z7-1 (Figure 1a–c). The growth and enzymatic production kinetic curve of strain Z7-1 revealed that it had excellent industrial advantages, including rapid growth, short enzyme production cycles, and long-term stable periods (Figure 1d,e). In a recent study, a KGM-degrading strain was screened and identified as *Bacillus amyloliquefaciens* WX-1, and this strain was shown to efficiently depolymerize KGM (1%) and produce a broad range of small-molecule oligosaccharides [14].

KGM is a large molecular polysaccharide that has high viscosity. For the industrial preparation of low-molecular-weight KGM or functional KGM oligosaccharides, it is crucial for an enzyme or microbial catalyst to maintain its degradation ability towards high-concentration KGM substrates during the degradation process. Currently, the substrate concentration of konjac powder is commonly controlled at 1% during biological degradation processes. Determining how to efficiently degrade high-concentration KGM accompanied by high viscosity is a huge challenge and technological bottleneck in the industrial process [12,13,14]. Here, strain Z7-1 showed good degradation ability for KGM at a high concentration of 5–20% (Figure 2), suggesting its advantage for KGM degradation in industrial conditions. These results showed that *B. licheniformis* Z7-1 might have an advantage for obtaining low-molecular-mass KGM or KGM oligosaccharides on a large scale over enzyme hydrolysis reported in previous findings [4,12,13,23,26]. In addition, *B. licheniformis* is a food-grade microorganism that has been widely used in various food fields such as traditional Chinese food and miso wine fermentation [34,35], and also as a qualified producer of β-cyclodextrin and amylase for industrial processes [36]. Moreover, *B. licheniformis* is considered to have great industrial potential for large-scale production of natural and recombinant enzymes [37,38,39]. 

KGM treated with *B. licheniformis* Z7-1 simultaneously generated many native oligosaccharides with or without acetyl groups. Natural non-acetylated oligosaccharides and single-acetylated oligosaccharides were detected in the Z7-1 hydrolysates (Figure 3b,c and Figure 5a). Traditional enzymatic hydrolysis of KGM using mannanase or cellulase can produce single-acetylated, double-acetylated, and triple-acetylated native oligosaccharides due to the degree of deacetylation of KGM materials and enzyme species during enzymatic hydrolysis [14]. Genome sequencing analysis of Z7-1 revealed that the carbohydrate-active enzyme CE4 or CE9 was responsible for polysaccharide deacetylation and at least seven polysaccharide deacetylases were present in the genome, which probably contributed to the reduction in the degree of deacetylation of the degraded KGM products.

We also observed that 1% KGM hydrolysates had a more lamellar structure and were smoother and flatter than those of 5% KGM, indicating that the degree of Z7-1 degradation was dependent on the concentration of KGM (Figure 5c). The reasonable explanation was that the high viscosity of konjac powder solution reduces the diffusion rate of bacteria, resulting in decreased enzyme production and degradation efficiency. Although the degradation efficiency of Z7-1 at 5% KGM was slightly lower than that at 1% KGM, Z7-1 maintained excellent hydrolysis efficiency at 5% KGM, indicating its industrial application. Microbial sources including bacteria and fungi have been explored for the secretion of enzymes to convert natural polysaccharides to functional oligosaccharides or low-molecular-mass components [13,16,18]. The amount and contents of extracellular enzymes from the microbial catalysis were generally affected by the composition of the culture medium, culture time, temperature, and substrate concentration, and these factors potentially contributed to the chemical and structural composition of degraded components and their physicochemical properties [14,16]. 

Antibiotic resistance to bacterial pathogens is a major concern for global human health and development. Resistance to common antibiotics is a pressing issue that affects global health and development. Therefore, efficient and safe alternatives to antibiotics are required to prevent and control drug-resistant pathogens. The KGM degradation products of strain Z7-1 exhibited antibacterial activity against various food-spoilage bacteria, including *B. cereus*, *P. fragi*, and *S. aureus* (Figure 3e), indicating their potential applications in food-spoilage prevention and clinical therapeutic strategies.

The monosaccharide composition of natural KGM and its degraded components revealed a reduction in the ratio of mannose to glucose after bacterial degradation, indicating that strain Z7-1 mainly utilizes glucose (Figure 3d). The preferential utilization of glucose by Z7-1 was consistent with the analysis of carbohydrate-active enzymes in the Z7-1 genome. Genome sequence analysis revealed that at least five cellulases exist in the genome of Z7-1 (Figure 6d). We also found that one gene coding lytic polysaccharide monooxygenases exists in the genome of Z7-1 (Figure 6d). Most bacteria-derived lytic polysaccharide monooxygenases belong to the AA10 family, which uses a new oxidative cleavage mechanism that differs from previous cellulases or chitinases and plays a critical role in degrading recalcitrant polysaccharides into fermentation sugars [40]. In a recent study, bacteria-derived *Sm*AA10A, which is chitin-active with strict C1 oxidation, was found to depolymerize KGM into low-molecular-weight KGM and KGM oligosaccharides, generating diverse oxidation products (e.g., single C1, single C4, and double C1/C4) [24]. In addition, the zymolytic analysis revealed the presence of at least five extracellular KGM-degrading enzymes, including three major and two minor enzymes secreted into the bacterial culture medium. The high thermal stability of these extracellular KGM-degrading enzymes also indicates their potential for industrial applications (Figure 7). The catalytic properties and mechanism of these extracellular enzymes responsible for KGM degradation will be further studied in the future.

## 5. Conclusions

Currently, industrial strains for KGM degradation are compromised by contamination from other bacteria and low hydrolysis efficiency, which impedes their application in high concentrations of KGM owing to its large molecular mass and high viscosity. Here, we identified the *B. licheniformis* Z7-1 strain from decaying konjac as manifesting an excellent degradation efficiency at high KGM substrate concentration (5–20%). The physicochemical properties of degraded KGM components were characterized, and the degradation products showed antibacterial activity against food-spoilage bacteria. Genome sequencing and zymolysis showed at least five extracellular carbohydrate-active enzymes with high thermal stability present in the genome of Z7-1. These results suggest that strain Z7-1 has potential for industrial application in the preparation of low-molecular-weight KGM products with antibacterial activity and provides an alternative method for converting natural biomass into functional components.

## Figures and Tables

**Figure 1 foods-13-02041-f001:**
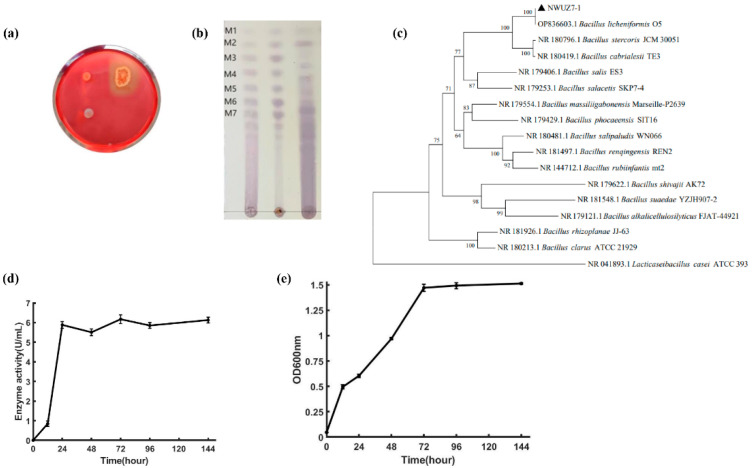
Isolation and characterization of KGM-degrading bacteria Z7-1. (**a**) Screening and isolation of KGM-degrading strain. (**b**) TLC analysis of KGM degradation products. (**c**) A phylogenetic tree based on 16S rDNA gene sequences of strain Z7-1 and other *Bacillus* species. Enzyme production (**d**) and growth kinetic curves (**e**) of strain Z7-1.

**Figure 2 foods-13-02041-f002:**
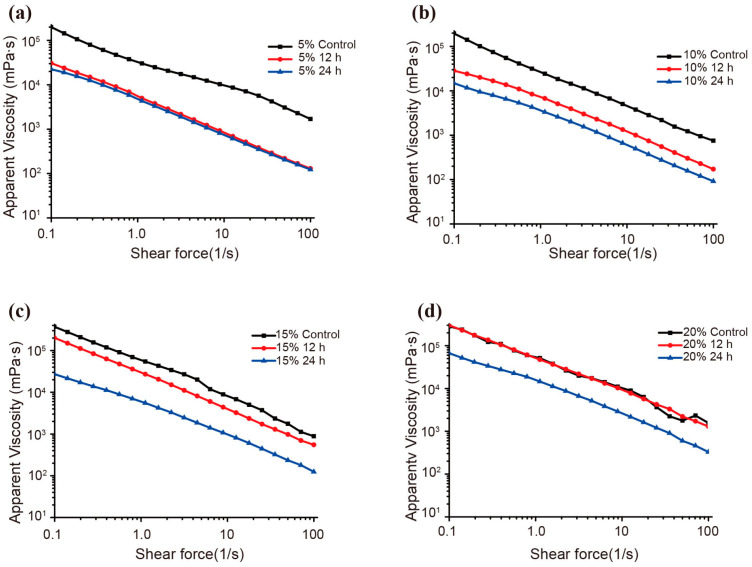
The apparent viscosity of KGM products degraded by strain Z7-1. Strain Z7-1 was cultured with 5% (**a**), 10% (**b**), 15% (**c**), or 20% (**d**) KGM before the apparent viscosity was determined after 12 h and 24 h.

**Figure 3 foods-13-02041-f003:**
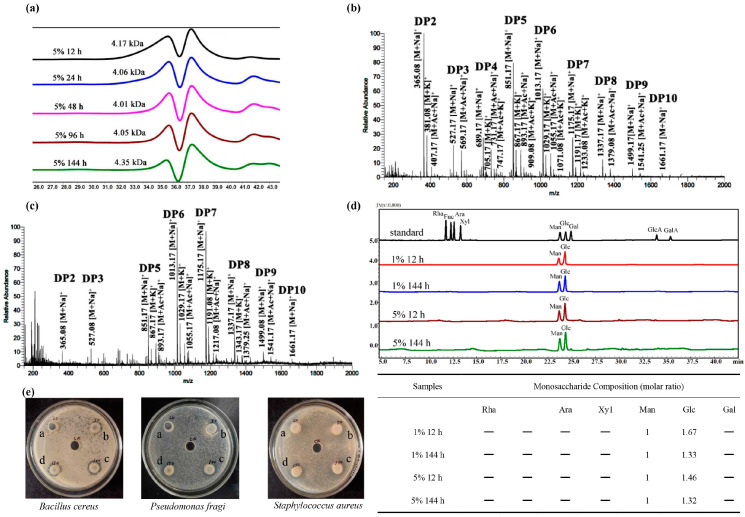
The physicochemical properties of products degraded by strain Z7-1. (**a**) The molecular mass of products degraded by strain Z7-1 acting on 5% KGM substrate for a series of different hydrolysis times (12 h, 24 h, 48 h, 96 h, and 144 h). ESI-MS analysis of products degraded by Z7-1 for 5% KGM substrate for 12 h (**b**) and 144 h (**c**). (**d**) The monosaccharide compositions analysis of products degraded by strain Z7-1 towards 1% and 5% KGM substrate concentration for 12 h and 144 h. (**e**) The effects of degradation products on food-spoilage bacteria such as *B. cereus*, *P. fragi*, and *S. aureus* (a, b, c, and d represent KGM degradation solutions after 24 h with volumes of 20, 50, 100, and 150 µL, respectively; CK is sterile water).

**Figure 4 foods-13-02041-f004:**
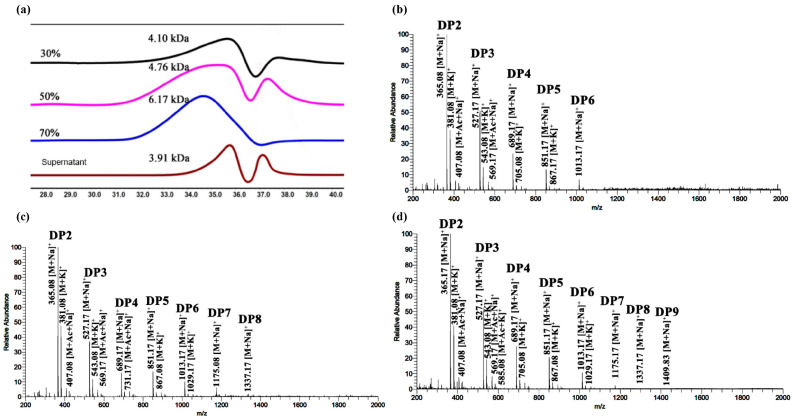
The molecular mass of KGM degradation components and the analysis of KGM oligosaccharides purified from a series of concentrations of alcohol precipitation. (**a**) The molecular masses of KGM degradation components were determined using high-performance liquid gel permeation chromatography (HPGPC). The composition of KGM oligosaccharides from 30% (**b**) and 50% (**c**) ethanol concentration precipitate and the supernatants (**d**) after 70% precipitation was determined by ESI-MS.

**Figure 5 foods-13-02041-f005:**
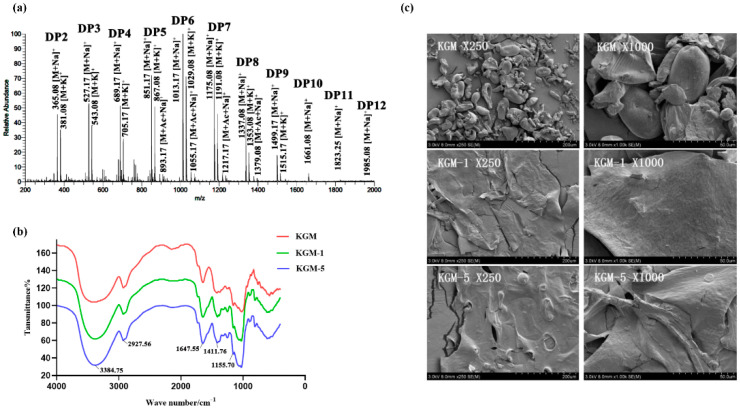
The physicochemical characterization of purified KGM degraded component. (**a**) ESI−MS analysis of KGM degraded component from 70% alcohol precipitation. (**b**) FT−IR of native KGM (1%) and degraded KGM components KGM−1 and KGM−5. KGM−1 and KGM−5: 70% precipitated component from the bacteria hydrolysates with the substrate concentration controlled at 1% (KGM-1) and 5% (KGM−5), respectively. (**c**) SEM micrographs of native KGM, KGM−1, and KGM−2 (magnification: 250× and 1000×).

**Figure 6 foods-13-02041-f006:**
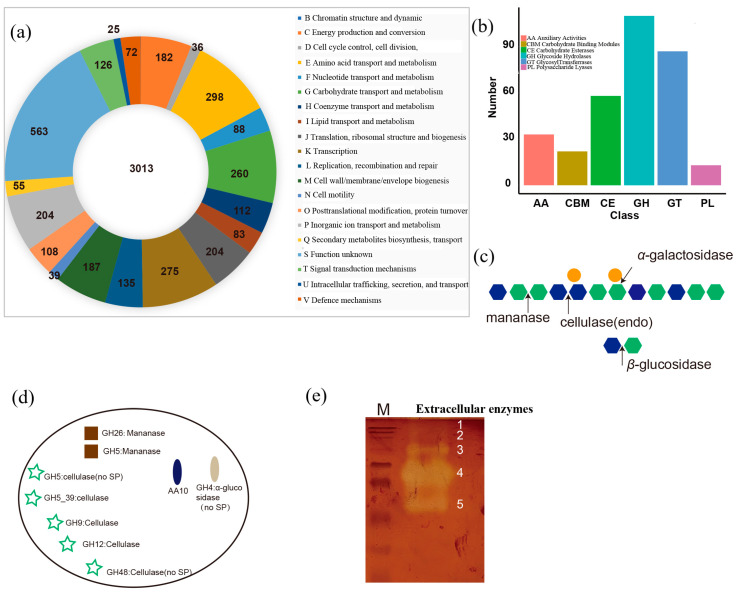
Functional categories in the Z7-1 genome and the carbohydrate-active enzymes that could contribute to KGM degradation. (**a**) Pie chart of the functional categories associated with the Z7-1 genome. Genes were annotated and categorized into clusters of orthologous groups (COGs), and the number of orthologous genes in each category is displayed. (**b**) Content of carbohydrate-active enzymes in the Z7-1 genome. The carbohydrate-active enzymes were clustered into six classes: auxiliary active (AA), carbohydrate-binding module (CBM), carbohydrate esterase (CE), glycoside hydrolases (GH), glycoside transferases (GT), and polysaccharides lysate (PL). (**c**) The composition and structures of KGM and the enzymes needed for its hydrolysis. (**d**) Enzymes including five cellulases, two mannanases, one α-galactosidases, and one additional AA10 are responsible for KGM degradation and appear in the genome of strain Z7-1 (SP represents signal peptides). (**e**) Zymogram analysis of extracellular enzymes in degrading KGM.

**Figure 7 foods-13-02041-f007:**
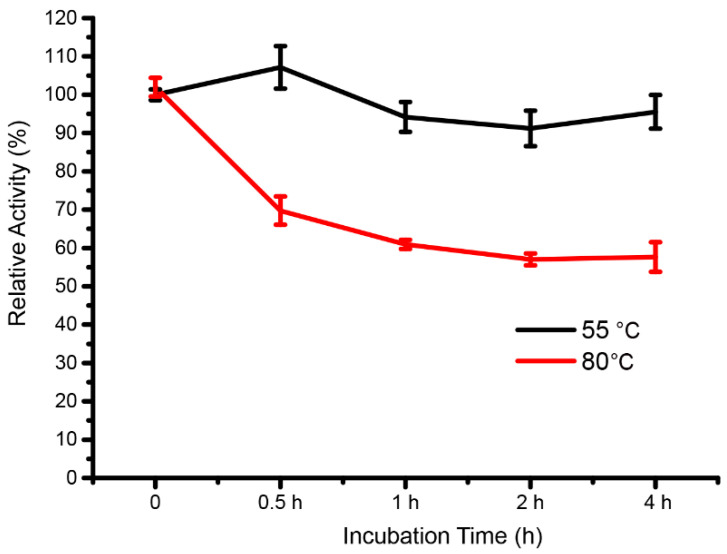
The thermal stability of extracellular enzymes.

## Data Availability

The original contributions presented in the study are included in the article, further inquiries can be directed to the corresponding author.

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
