# Peer review of "Characterization of Degraded Konjac Glucomannan from an Isolated Bacillus licheniformis Strain with Multi-Enzyme Synergetic Action"

_foods, 2024, doi:10.3390/foods13132041_

Round 1
Reviewer 1 Report
Comments and Suggestions for Authors
This is a relevant study that isolated and investigated the efficiency and effects of Bacillus licheniformis Z7-1 strain on the degradation of Konjac Glucomannan (KGM) as well as the characteristics and properties of its degradation products.
Some considerations:
1) I suggest that the Authors complement the “Materials and Methods” section with sufficiently detailed information to allow a clear understanding and reproducibility of the study. Additionally, add information about the quantification of bacteria, KGM concentrations and the parameters and experimental conditions used in the study;
2) Describe more clearly the results regarding the analysis of the composition of KGM degradation products. Add the data referring to Sfigs 1, 2 and 3 mentioned in the manuscript;
3) I suggest that the Authors synthesize information already mentioned or described in previous sections. Considering data available in the literature, discuss more broadly aspects related to the optimization of effective KGM degradation methods to expand and improve their applicability, as well as the impact of Z7-1 strain activity on the chemical and structural composition of KGM and its physicochemical properties and biological, better addressing biochemical and molecular changes.
Author Response
|
2. Point-by-point response to Comments and Suggestions for Authors |
|
Comments 1: I suggest that the Authors complement the “Materials and Methods” section with sufficiently detailed information to allow a clear understanding and reproducibility of the study. Additionally, add information about the quantification of bacteria, KGM concentrations and the parameters and experimental conditions used in the study. |
|
Response 1: We feel great thanks for your professional suggestions on our work. As the reviewer’s advice, we have complemented the “Materials and Methods” section with sufficiently detailed information about the experiment process and quantification of bacteria, KGM concentrations, and the parameters and experimental conditions used in the study in the revised manuscript. Please see the section of “Materials and Methods”, Lines 142-153, Lines 163-173, Lines 178-185, Lines 205-211 in the revised manuscript. Added information: Lines 142-153 as follows: Strain Z7-1 obtained were inoculated into 20 mL Luria-Bertani broth medium in a 50 mL conical flask and cultured at 37 ℃ for 12 h with shaking at 180 rpm. 2 mL of cultures was transferred into 100 mL of Luria-Bertani broth medium, and then was cultured for 48 h at 37 ℃ before different concentrations of 5%, 10%, 15%, and 20% konjac flour were added to the cultures. Samples of the bacterial hydrolysis products were collected at 12 h and 24 h, and the changes in viscosity were measured using an Anton Paar MCR 302 rheometer. A power-law model was used to fit the viscosity of each sample to the shear rate. τ = K×γ n, where τ represents shear stress, K is the consistency coefficient or power law coefficient (Pa⋅sn), and n represents the fluidity index or power law index (unitless). The K value is a measure of viscosity but is not equal to the viscosity value. The higher the viscosity, the higher the K value. Lines 163-173: The mobile phase contained phosphate buffer (20 mM, pH 6.0) with a flow rate of 0.3 mL/min. Monosaccharide composition analysis of the degraded KGM components was performed as previously described [24]. Four milligrams of the sample were hydrolyzed by trifluoroacetic acid at 121℃ for 2 h in a reaction flask, and then the pH of the hydrolysate was adjusted to neutral. NaBH4 (w/v, 4%, 0.5 mL) was added and allowed to react at room temperature for 1.5 h. Excess NaBH4 was removed by adding glacial acetic acid and methanol. 1 mL of pyridine and acetic anhydride was added into the reaction system to perform the acetylation, followed by incubation overnight at room temperature. The targeted sample was dried under vacuum at 80 ℃ and further extracted using dichloromethane and double distilled water. Lines 178-185: First, 4 mg of the sample was dissolved in 6 mL water and loaded into a Sep-Pak C18 solid-phase extraction column, which was pre-balanced with water. The target glycans were eluted with 12 mL of water and then loaded onto a pre-balanced graphitic‑carbon solid-phase extraction column and washed with 30 mL water to remove unbound components. The bound glycans were eluted with 25 % acetonitrile in water, and the targeted oligosaccharides were dried under a stream of nitrogen at room temperature. Lines 205-211: Cells were removed by centrifugation at 12,000 g for 10 min, and the supernatant samples were used for the antibacterial activity assays. An Oxford cup (inner diameter, 7.8 mm; Xuzhou Xinri Technology, Xuzhou, China) was filled with the supernatant samples at a volume of 20, 50, 100, and 150 µL; and 150 µL sterile water was used as the control. Then the set-up was incubated at 37 ℃ for 36 h. The evaluation of antimicrobial activities was performed based on clear zones developed around the wells. |
|
Comments 2: Describe more clearly the results regarding the analysis of the composition of KGM degradation products. Add the data referring to Sfigs 1, 2 and 3 mentioned in the manuscript; |
|
Response 2: Thanks for the reviewer’s careful reading of our manuscript. According to the reviewer’s advice, we have added Sfigs 1, 2 and 3 mentioned in the manuscript as the Fig.4 in the revised manuscript. Please see P9 Lines 337-344 in the new version. |
|
Comments 3: I suggest that the Authors synthesize information already mentioned or described in previous sections. Considering data available in the literature, discuss more broadly aspects related to the optimization of effective KGM degradation methods to expand and improve their applicability, as well as the impact of Z7-1 strain activity on the chemical and structural composition of KGM and its physicochemical properties and biological, better addressing biochemical and molecular changes. |
|
Point 1: |
|
Response 1: Thanks for the reviewer’s professional advice. According to the reviewer’s advice, we have discussed more broadly aspect the optimization of effective KGM degradation methods and the impact of Z7-1 strain activity on the chemical and structural composition of KGM and its physicochemical properties in the revised version. The added information was as follows: Microbial sources including bacteria and fungi have been explored for the secretion of enzymes to convert the natural polysaccharides to functional oligosaccharides or low-molecular-mass components [13,16,18]. The amount and contents of extracellular enzymes from the microbial catalysis were generally affected by the composition of the culture medium, culture time, temperature, and substrate concentration; and this factors potentially contributed to the chemical and structural composition of degraded components and its physicochemical properties [14,16]. Please see the discussion section Lines 493-499 in the revised version. |
Reviewer 2 Report
Comments and Suggestions for Authors
The manuscript presents an interesting topic on the degraded konjac glucomannan from an isolated strain with multi-enzyme synergetic action.
I recommend that the authors should change the repeated keywords in the title.
In lines 145-146, the authors should describe so well the rheological test to measure the apparent viscosity. Also, ESI-MS analysis should be improved to include the complete methodology.
The authors include only one-time abbreviations in the manuscript.
Figure 2 should be improved. The correct axes should be apparent viscosity vs shear rate. Also, I recommend that the authors should use a mathematical model to calculate the rheological parameters of the behavior presented by the samples.
In the figure 3a, the authors should present the statistical analysis of the molecular weight of the samples. The chromatograms and values are very similar.
How can the authors demonstrate the statistical difference in the antimicrobial activity?
I recommend that the authors improve the quality of Figures 3b, 3c, and 3d. Also, I recommend that they include a table with the identification of molecules by ESI-MS.
Finally, was MS spectroscopy coupled to chromatographic systems such as UPLC or HPLC
Author Response
|
1. Summary |
|
|
|
Thank you very much for taking the time to review this manuscript. Please find the detailed responses below and the corresponding revisions/corrections highlighted/in track changes in the re-submitted files. |
||
|
Comments 1: The manuscript presents an interesting topic on the degraded konjac glucomannan from an isolated strain with multi-enzyme synergetic action. I recommend that the authors should change the repeated keywords in the title. |
||
|
Response 1: Thanks for the reviewer’s professional advices. We have modified the title as follows: Characterization of Degraded Konjac Glucomannan from an Isolated Bacillus licheniformis Strain with Multi-Enzyme Synergetic Action. |
||
|
Comments 2: In lines 145-146, the authors should describe so well the rheological test to measure the apparent viscosity. Also, ESI-MS analysis should be improved to include the complete methodology. |
||
|
Response 2: Thanks for the reviewer’s advice, according to the reviewer’s suggestion, we have described the rheological test to measure the apparent viscosity as follows: Samples of the bacterial hydrolysis products were collected at 12 h and 24 h, and the changes in viscosity were measured using an Anton Paar MCR 302 rheometer. A power-law model was used to fit the viscosity of each sample to the shear rate. τ = K×γ n, where τ represents shear stress, K is the consistency coefficient or power law coefficient (Pa⋅sn), and n represents the fluidity index or power law index (unitless). The K value is a measure of viscosity but is not equal to the viscosity value. The higher the viscosity, the higher the K value. In addition, the detailed information about the ESI-MS analysis has been improved in the revised manuscript as follows: The targeted oligosaccharides were dried under a stream of nitrogen at room temperature. Identification and analysis of the KGM oligosaccharides were performed using ESI-MS (LTQ-Tune; Thermo Scientific) as follows: 2 μL of the sample solutions were injected into the electrospray ion source by a stream of 50% methanol at a flow rate of 20 μL·min−1 from the pump of HPLC system. For the electrospray ion source, the spray voltage was set at 4 kV, with a sheath nitrogen flow rate of 20 arb, an auxiliary gas (nitrogen) flow rate of 10 arb, temperature at 300 °C, voltage at 37 V, and a tube lens voltage of 250 V. |
||
|
Comments 3: The authors include only one-time abbreviations in the manuscript. Response 3: Thanks for the reviewer’s careful advice and we are so sorry for our carelessness. We have checked the thorough manuscript and modified it in the revised version. Comments 4: Figure 2 should be improved. The correct axes should be apparent viscosity vs shear rate. Also, I recommend that the authors should use a mathematical model to calculate the rheological parameters of the behavior presented by the samples. Response 4: We are grateful for the reviewer’s professional and kind suggestion. According to the reviewer’s advice, we have modified the axes as the apparent viscosity vs shear rate in the new version. In addition, we have added the related information of a mathematical model to calculate the rheological parameters of the behavior presented by the samples in the revised manuscript. A power-law model was used to fit the viscosity of each sample to the shear rate. τ = K×γ n, where τ represents shear stress, K is the consistency coefficient or power law coefficient (Pa⋅sn), and n represents the fluidity index or power law index (unitless). The K value is a measure of viscosity but is not equal to the viscosity value. The higher the viscosity, the higher the K value. |
||
|
Comments 5: In the figure 3a, the authors should present the statistical analysis of the molecular weight of the samples. The chromatograms and values are very similar. |
||
|
Response 5: we are grateful for the reviewer’s suggestion. According to the reviewer’s advice, we have added the statistical analysis of the molecular weight of the samples of Fig.3a in the manuscript as follows: As shown in Fig. 3a, after degradation of 12 h or 24 h, the degraded products (4.17±0.05 kDa for 12 h and 4.06±0.09 kDa for 24 h) presented wider peaks in the GPC profile, indicating that the components are probably composed of a series of polymers with different molecular weights. When the time extension to 48 h (4.01±0.07 kDa), 96 h (4.05±0.02 kDa), and 144 h (4.35±0.08 kDa). Please see the P9, Lines 289-293 in the revised version. Comments 6: How can the authors demonstrate the statistical difference in the antimicrobial activity? Responses 6: Thanks for the reviewer’s professional advice. The antibacterial activity was evaluated by the obvious antibacterial zones on the LB solid agar and the diameters of the clear zones recorded. We have added the statistical difference in the antimicrobial activity in the revised manuscript as follows: Larger antibacterial circles could be observed with the addition of more KGM degradation products into the cups (Fig. 3e), and the diameters of clear zones were as follows: B. cereus (a, 8.01±0.02 mm; b, 8.53±0.01 mm; c, 9.01±0.04 mm; d, 9.24±0.05 mm), P. fragi (a, 7.90±0.03 mm; b, 8.61±0.03 mm; c, 9.51±0.08 mm; d, 9.98±0.02 mm), S. aureus (a, 8.01±0.03 mm; b, 8.63±0.01 mm; c, 9.60±0.10 mm; d, 10.18±0.05 mm). Please see P9, Lines 329-335 in the revised version. Comments 7: I recommend that the authors improve the quality of Figures 3b, 3c, and 3d. Also, I recommend that they include a table with the identification of molecules by ESI-MS. Finally, was MS spectroscopy coupled to chromatographic systems such as UPLC or HPLC. Responses 7: We deeply appreciate the reviewer’s pertinent and constructive advice. In this present study, our work mainly focuses on screening the target strain and characterization of the degraded KGM product. The DPs of KGM-oligosaccharides of degraded products at different times intervals were analyzed by ESI-MS. The results mainly elucidate the preference of strain Z7-1 on the DPs oligosaccharides during degradation. In addition, MS spectroscopy was coupled to chromatographic systems HPLC and we have added the related information of ESI-MS methodology in the “Material and Method” in the revised manuscript. We are extremely grateful for the reviewer to give the constructive suggestion again and we will consider the reviewer’s constructive suggestion in our next work. |
||